# Oleuropein Ameliorates Advanced Stage of Type 2 Diabetes in *db*/*db* Mice by Regulating Gut Microbiota

**DOI:** 10.3390/nu13072131

**Published:** 2021-06-22

**Authors:** Shujuan Zheng, Yanan Wang, Jingjing Fang, Ruixuan Geng, Mengjie Li, Yuhan Zhao, Seong-Gook Kang, Kunlun Huang, Tao Tong

**Affiliations:** 1Beijing Advanced Innovation Center for Food Nutrition and Human Health, College of Food Science and Nutritional Engineering, China Agricultural University, Beijing 100083, China; zhengshujuan1015@163.com (S.Z.); wangyanan_ytu@126.com (Y.W.); fangjingjing@cau.edu.cn (J.F.); 17768128861@163.com (R.G.); mengjieli1112@163.com (M.L.); 13120164842@163.com (Y.Z.); 2Department of Food Engineering, Mokpo National University, Muangun 58554, Korea; sgkang@mokpo.ac.kr; 3Key Laboratory of Safety Assessment of Genetically Modified Organism (Food Safety), Ministry of Agriculture, Beijing 100083, China

**Keywords:** oleuropein, advanced stage of type 2 diabetes, glucose tolerance, gut microbiota

## Abstract

Previous studies have reported the therapeutic effects of oleuropein (OP) consumption on the early stage of type 2 diabetes. However, the efficacy of OP on the advanced stage of type 2 diabetes has not been investigated, and the relationship between OP and intestinal flora has not been studied. Therefore, in this study, to explore the relieving effects of OP intake on the advanced stage of type 2 diabetes and the regulatory effects of OP on intestinal microbes, diabetic *db*/*db* mice (17-week-old) were treated with OP at the dose of 200 mg/kg for 15 weeks. We found that OP has a significant effect in decreasing fasting blood glucose levels, improving glucose tolerance, lowering the homeostasis model assessment–insulin resistance index, restoring histopathological features of tissues, and promoting hepatic protein kinase B activation in *db*/*db* mice. Notably, OP modulates gut microbiota at phylum level, increases the relative abundance of *Verrucomicrobia* and *Deferribacteres*, and decreases the relative abundance of Bacteroidetes. OP treatment increases the relative abundance of *Akkermansia*, as well as decreases the relative abundance of *Prevotella*, *Odoribacter*, *Ruminococcus*, and *Parabacteroides* at genus level. In conclusion, OP may ameliorate the advanced stage of type 2 diabetes through modulating the composition and function of gut microbiota. Our findings provide a promising therapeutic approach for the treatment of advanced stage type 2 diabetes.

## 1. Introduction

Diabetes mellitus is a chronic metabolic condition characterized by hyperglycemia caused primarily by insulin resistance, inadequate insulin production, or both [1]. Diabetic patients with type 2 diabetes account for roughly 90% of all people with diabetes. Reduced insulin sensitivity in critical organs such as adipose tissues, muscle, and liver is a hallmark of type 2 diabetes. In 2019, the International Diabetes Federation reported 463 million diabetic patients globally, accounting for 9.3% of the global population; with the rising incidence, it is anticipated that by 2030, there will be 578 million diabetics globally. As the number of people diagnosed with type 2 diabetes rises, more effective drugs are needed to relieve type 2 diabetes in various patients. Progression to diabetes can be viewed as having definable stages characterized by changes in various metabolic parameters and β-cell function, mass, and phenotype [2]. For example, there was a significant difference in glucose-stimulated insulin secretion between an early and advanced stage of diabetes [3]. Compensatory hypertrophy of the islets can be seen in the early stage of diabetes, but the capacity of such compensation is markedly attenuated in the advanced stage in obese type 2 diabetes model *db*/*db* mice exhibiting significant hyperglycemia [4]. In addition, in the advanced or late stage of diabetes, several complications arise such as diabetic neuropathy, diabetic retinopathy, diabetes cardiovascular complications, and diabetic nephropathy.

Gut microbes can influence several physiological processes by interacting with the host, according to mounting evidence. Dysregulation in the composition and activity of gut microorganisms may create metabolic abnormalities in the host, leading to diabetes and other metabolic disorders. It has been established that a variety of factors, including lifestyle, genotype, and nutrition, alter the host’s health via regulating intestinal microorganisms [5]. Various edible natural functional ingredients can alleviate type 2 diabetes by reshaping intestinal microbes [6]. As a result, controlling gut bacteria could become a viable treatment option for type 2 diabetes.

Polyphenols are premium natural phytochemicals for the treatment of diabetes owing to their multi-target efficacies, including improving liver glucose homeostasis, promoting insulin secretion, inhibiting glucose transport, attenuating the activities of α-amylase and α-glucosidase, and remodeling intestinal microbes [1]. Oleuropein (OP), which exists in olive leaves, unripe olive fruits, olive branches, and olive oil, is a highly bioactive phenolic compound with various biological functions [7,8]. Previous studies have shown that OP consumption exerts therapeutic effects on the early stage of type 2 diabetes by repairing islet morphology, regulating insulin secretion, and improving glucose tolerance and insulin sensitivity [9,10,11,12]. However, the alleviating effects of OP on the advanced stage of type 2 diabetes and the regulatory effects of OP on intestinal microbes have not yet been investigated.

This study demonstrates that OP treatment is a practical approach to ameliorating the advanced stage of type 2 diabetes in *db*/*db* mice, which is related to the modulation of the gut microbiota. Our results provide a novel potential option for the management of the advanced stage of type 2 diabetes.

## 2. Materials and Methods

### 2.1. Materials

OP (analytical regent grade) was purchased from Chengdu purify Technology Development Co. Ltd. (Chengdu, China). D-(+)-glucose was obtained from Sigma (St. Louis, MO, USA). Insulin (Protamine Biosynthetic Human Insulin Injection (pre-mixed 30R), 100 IU/mL × 3 mL) was purchased from Novo Nordisk (Copenhagen, Denmark). Primary antibodies against β-tubulin (cat.# 2146), protein kinase B (Akt) (cat.# 4691S), and phospho-Akt (Thr 308) (cat.# 13038S) were purchased from Cell Signaling Technology (Danvers, MA, USA). The secondary antibody (cat.# A0208) was purchased from Beytime (Shanghai, China).

### 2.2. Animals and Designs

China Agricultural University Laboratory Animal Welfare and Animal Experimental Ethical committee approved the animal experiment protocol (AW11099102-3-4, Beijing, China). Eight-week-old BKS-*Lepr^em2Cd479^*/Gpt (*db*/*db*) male mice (*n* = 15) and wild type BKS-DB (*m*/*m*) male mice (*n* = 7) were obtained from Gempharmatech Co., Ltd. (Nanjing, China) and housed in a specific pathogen-free animal room of Animal Centre (SYXK (Jing) 2020-0052) with a 12 h daylight cycle (lights on at 6:30 a.m.). The room humidity range is 40–70% and the temperature range is 22 ± 2 °C. Studies showed that 16-week-old *db*/*db* mice exhibited both high fasting blood glucose and casual blood glucose, and were considered a model of advanced stage of type 2 diabetes [3,13]. Standard chow diet (Beijing Huafukang Biotechnology Co., Ltd., Beijing, China) was given to all the mice *ad libitum* until they were 17 weeks old. Then, the *db*/*db* mice were assigned to two groups: *db*/*db* group (*n* = 7) and *db*/*db* + OP group (*n* = 8). During the experiment period, animals were still fed with the standard chow diet and housed 3–4 per cage, with free access to water. OP (200 mg/kg) or vehicle (water) was daily orally gavaged to the *db*/*db* mice. During the treatment period, body weight was weighed weekly.

After 15 weeks of treatment, blood samples were obtained from the posterior orbital venous plexus. Then, the mice were dissected. The pancreas, epididymal adipose tissue, liver, and heart were weighed, snap-frozen in liquid nitrogen, and saved at −80 °C. A portion of these tissues was immersed in 4% paraformaldehyde solution. The serum was prepared by centrifugation (2500× *g* rpm, 15 min) and saved at −80 °C for subsequent experiments.

### 2.3. Fasting Blood Glucose and Oral Glucose Tolerance Test (OGTT)

During the experimental period, we measured the fasting blood glucose levels of the *db*/*db* and *db*/*db* + OP group mice at week 0, 1, 2, 3, 4, 9, and 15 by using Green Doctor (Green Cross Corporation, Yongin, South Korea) glucometer and glucose strips from the tail vein after 6 h fasting (8:00 a.m. to 2:00 p.m.). After 8 weeks of OP treatment, OGTT was performed in mice. Briefly, the OGTT was executed after fasting for 6 h (8:00 a.m. to 2:00 p.m.), after which the mice were orally gavaged 0.75 g/kg glucose. Blood samples were collected from the tip of the tail vein. Blood glucose was determined by the glucose meter before and 15, 30, 60, 90, and 120 min after the mice orally gavaged glucose.

### 2.4. Insulin Tolerance Test (ITT)

After 14 weeks of OP treatment, ITT was performed in mice following 6 h of food deprivation (8:00 a.m. to 2:00 p.m.). The mice were injected intraperitoneally with insulin (2 U/kg for *db*/*db* and *db*/*db* + OP group mice and 0.75 U/kg body weight for *m*/*m* group mice). Blood glucose was measured as described in OGTT.

### 2.5. Acute Intervention

Acute interventions were performed when the other *db*/*db* mice (*n* = 6) were 49 weeks old. Initial blood glucose was measured between 8:00 a.m. and 9:00 a.m. after light phase onset (6:30 a.m.) at random. Then, 200 mg/kg OP or vehicle (water) was administered orally, and the second blood glucose was measured after fasting for 4 h.

### 2.6. Biochemical Analysis

The levels of alanine aminotransferase (ALT), aspartate aminotransferase (AST), low-density lipoprotein cholesterol (LDL-C), high-density lipoprotein cholesterol (HDL-C), total cholesterol (TC), and total triglyceride (TG) in serum samples were analyzed by using BS-350E Mindray automatic biochemical analyzer (Shenzhen Mindray Biomedical Electronics Co., Ltd., Shenzhen, China).

The levels of fasting serum insulin and GLP-1 were determined by the Mouse Insulin ELISA kit (GSB-E05071m, Cusabio, Wuhan, China) and mouse glucagon-like peptide-1 (GLP-1) ELISA kit (GSB-E08118m, Cusabio, Wuhan, China), respectively. Homeostasis model assessment–insulin resistance (HOMA-IR) was calculated by the formula below:HOMA-IR = insulin (μIU/mL) × glucose (mmol/L)/22.5

### 2.7. Histology Staining

Tissues (liver, pancreas, and heart) fixed in 4% paraformaldehyde solution were dehydrated by gradient ethanol, cleared in xylene, wax sealed, and sectioned into 4.5 μm sections. Hematoxylin and eosin (H&E) were used to stain all specimens. Finally, neutral resin was applied to seal all samples. Histopathological changes were observed under bright field using a Leica DM750 microscope (Leica, Nussloch, Germany).

### 2.8. Real-Time Quantitative PCR

The liver RNA was extracted by Trizol reagent (TIANGEN, Beijing, China). The RNA concentration was measured with a Nano 300 spectrophotometer (Allsheng Co., Ltd., Hangzhou, China). The cDNA was synthesized by TransScript One-Step gDNA Removal and cDNA Synthesis SuperMix (TransGen Biotech, Beijing, China). Real-time quantitative PCR (RT-qPCR) was performed with SuperReal PreMix Plus (SYBR Green) (TransGen Biotech, Beijing, China). Primer sequences for qPCR are displayed in Table 1. The real-time PCR amplifications were performed as follows: 95 °C for 3 min, 40 cycles of 95 °C for 10 s and 55 °C for 30 s. Expression of β-actin was used to normalize the mRNA expression.

### 2.9. Western Blot Analysis

RIPA buffer (Solarbio, Beijing, China) was used to extract proteins of the liver. BCA protein assay kit (Solarbio, Beijing, China) was used to determine the protein concentration. The hepatic proteins were separated by 10% sodium dodecyl sulfate polyacrylamide gel electrophoresis (SDS-PAGE) and transferred onto the poly (vinylidene fluoride) (PVDF) membrane. Bovine serum albumin (5%) in TBST was used to block the membranes at room temperature for 1 h. After blocking, the membranes were incubated with the primary antibodies at 4 °C overnight. Then, the membranes were washed three times. The secondary antibody was used to incubate the membranes at room temperature for 1 h. Protein bands were visualized using chemiluminescent HRP substrate (Millipore Corporation, Billerica, USA), and all signals were visualized and analyzed by Clinx ChemiCapture 3300 (Clinx Science Instruments Co., Ltd., Shanghai, China). The expression levels of proteins were quantified using ImageJ version 1.8.0 (National Institutes of Health, Bethesda, MD, USA).

### 2.10. Gut Microbiota Analysis

The contents of the cecum were collected and flash-frozen in liquid nitrogen before being stored at -80°C. The microbial DNA from cecal contents was extracted using the methods reported previously [14]. The purity of DNA was detected by Nano-300 micro spectrophotometer. Using a 1% agarose gel electrophoresis, the quality of DNA was determined.

At Novogene Bio-informatics Institute, the V3-V4 region of the 16S rRNA gene was amplified by PCR and sequenced on an Illumina Nova-PE250 platform (San Diego, CA, USA) (Beijing, China). Quantitative Insights into Microbial Ecology (QIIME) version 1.9.1. (Caporado lab—Northern Arizona University, Flagstaff, AZ, USA) was used to analyze the sequences. The same operational taxonomic unit (OTU) was allocated to sequences that were ≥97% identical. Ribosomal Database Project (RDP) version 2.2 classifier was used to perform taxonomic annotation [15]. Receiver operating characteristics (ROC) curve was plotted with the use of GraphPad Prism version 8 (GraphPad Software Inc, San Diego, CA, USA). Nonmetric multidimensional scaling (NMDS) plot and alpha diversity were analyzed by using Paleontological Statistics (PAST) version 3.22 [16]. The present features were subjected to linear discriminant analysis (LDA) effect size (LEfSe) (http://huttenhower.sph.harvard.edu/galaxy/ (accessed on 11 May 2021)) followed by LDA analysis to measure the size effect of each abundant taxon, and two filters (p 0.05 and LDA score > 3) to identify bacterial taxa that were differentially represented between groups at the genus or higher taxonomic levels. The Spearman’s rho non-parametric correlations between gut microbiota at species level and fasting blood glucose, blood glucose at time point of 15 min during OGTT, and HOMA-IR of *db*/*db* mice were then calculated by using SPSS version 23.0 (IBM, Armonk, NY, USA). The heatmap was created using GraphPad Prism program. The function profiles of the gut microbiota were predicted using Phylogenetic Investigation of Communities by Reconstruction of Unobserved States (PICRUSt) (http://huttenhower.sph.harvard.edu/galaxy/ (accessed on 13 May 2021)), and statistical significance differences were calculated using Statistical Analyses of Metagenomic Profiles (STAMP) version 2.1.3 (http://kiwi.cs.dal.ca/Software/STAMP (accessed on 15 May 2021)).

### 2.11. Statistical Analysis

All data were analyzed with GraphPad Prism version 8 and represented as mean ± Standard Error of Mean (SEM). We used an unpaired two-tailed Student’s *t*-test for comparisons, and *p* < 0.05 was considered statistically significant.

## 3. Results

### 3.1. OP Alleviated Advanced Stage of Type 2 Diabetes in db/db Mice

We investigated 17-week-old *db*/*db* mice to see if OP could help with advanced type 2 diabetes. The findings revealed that starting at week 9, OP significantly lowered fasting blood glucose in *db*/*db* mice, and this hypoglycemic impact was maintained until week 15 (Figure 1a). In the OGTT assay, OP treatment markedly decreased blood glucose levels at time 0, 15, and 90 min and the area under the curve (AUC) in *db*/*db* mice (Figure 1b). OP treatment did not alter insulin sensitivity or body weight in *db*/*db* mice (Figure 1c,d). The fasting serum insulin levels in *db*/*db* + OP group mice tended to be lower compared with those of *db*/*db* group mice, although this did not reach statistical significance (Figure 1e). OP significantly reduced the HOMA-IR index in *db*/*db* mice (Figure 1f). OP treatment did not change the fasting serum GLP-1 levels (Figure 1g). To determine whether the effect of OP in lowering *db*/*db* fasting glucose was related to acute stimulation of insulin secretion, we performed an acute glucose experiment. After 4 h of single administration of OP to *db*/*db* mice, we did not observe an acute ameliorative effect of OP on blood glucose (Figure 1h). OP treatment did not significantly alter the levels of AST, ALT, TG, TC, LDL-C, and HDL-C in the serum of *db*/*db* mice (Table 2).

### 3.2. OP Treatment Restored Histopathological Features of Tissues in db/db Mice

The *db*/*db* group mice had similar pancreas mass, higher liver mass, and higher epididymal fat mass compared to *m*/*m* mice. For tissue coefficients, compared to *m*/*m* group mice, *db*/*db* group mice had significantly lower pancreatic coefficients, unchanged liver coefficients, and significantly higher epididymal fat coefficients. OP treatment did not affect pancreas weight, pancreas coefficients, liver weight, liver coefficients, and epididymal fat weight in *db*/*db* mice. OP significantly decreased epididymal fat coefficients (Table 3).

Via observing H&E-stained pathological sections through a microscope, we found that OP restored histopathological features of tissues in *db*/*db* mice. Islet damage is a kind of typical feature of type 2 diabetes. Our results showed that the islet area of *db*/*db* mice was smaller than that of *m*/*m* mice. OP treatment increased the islet area of *db*/*db* mice (Figure 2a). The microscopic photograph of the pathological section showed that the liver tissue of *db*/*db* mice had lipid deposition and disordered arrangement of hepatocytes. OP reduced hepatic lipid deposition and enabled an orderly arrangement of hepatocytes in *db*/*db* mice (Figure 2b). Mice in the *m*/*m* group had neatly arranged myocardial fibers, while mice in the *db*/*db* group had disordered myocardial fibers. OP treatment resulted in more neatly arranged myocytes in *db*/*db* mice (Figure 2c). Overall, these results indicated that OP treatment alleviated pancreatic, liver, and heart damage in *db*/*db* mice.

### 3.3. The Effects of OP on Hepatic Insulin Signaling and Key Enzymes Related to Hepatic Gluconeogenesis in db/db Mice

The protein tyrosine phosphatase 1B (PTP1B) negatively modulates the insulin signaling pathway by inhibiting the phosphorylation of insulin receptors and insulin receptor substrates. Our experimental results showed that OP was able to inhibit the hepatic *PTP1B* mRNA expression in *db*/*db* mice, which indicated that OP treatment may affect the insulin signaling pathway (Figure 3a). In order to further investigate the effect of OP on the hepatic insulin signaling pathway, we measured the phosphorylation of Akt. We found that OP increased the level of Akt phosphorylation in the liver of *db*/*db* mice, which signifies that OP activates the hepatic insulin signaling pathway in *db*/*db* mice (Figure 3b). We also determined the mRNA expressions of three key enzymes of hepatic gluconeogenesis. The results showed that OP treatment significantly reduced the fructose-1,6-bisphosphatase-1 (*Fbp1*) mRNA expression, while OP had no effect on the mRNA expression of phosphoenolpyruvate carboxykinase (*P**epck*) and glucose-6-phosphatase (*G6pase*) (Figure 3c–e). Our experimental results showed that OP did not alter liver glycogen levels of *db*/*db* mice (Figure 3f).

### 3.4. The Effects of OP Treatment on α-Diversity and β-Diversity of Gut Microbiota in db/db Mice

Next, we determined the effects of OP on the regulation of intestinal flora in *db*/*db* mice. We collected cecum contents from *m*/*m* group, *db*/*db* group, and *db*/*db* + OP group mice and extracted the gut microbial genomes of these animals. The number of OTUs in *db*/*db* group mice was lower than that in *m*/*m* group mice, but OP treatment increased the OTU number of *db*/*db* mice (Figure 4a). To determine whether the sequencing results obtained could be used to distinguish among these three groups of mice, we plotted ROC curves. The area under the ROC curve was found to be 0.6246, so the sequencing results obtained could be used to distinguish among these three groups of mice (Figure 4b). β-Diversity measures the diversity between inter community. NMDS based on the OTU abundance matrix demonstrated that the β-diversity of gut microbial communities was markedly different among the three groups and the microbial structure was influenced by OP consumption in *db*/*db* mice (Figure 4c). α-Diversity represents the evenness and richness of the intestinal microbiota within the sample. At the OTU level, indices of α-diversity such as Simpson, Shannon, Chao1, Dominance, and Brillouin were measured. The Chao1 index was significantly different between *m*/*m* group mice and *db*/*db* group mice. Simpson, Shannon, Dominance, and Brillouin indices were significantly different between *db*/*db* group mice and *db*/*db* + OP group mice (Figure 4d–h).

### 3.5. OP Treatment Changed the Gut Microbiota Composition in db/db Mice

The composition and relative abundance of gut microbiota at the phylum and genus levels in the three groups of mice are shown in Figure 5. The OP treatment decreased the relative abundance of Bacteroidetes and increased the relative abundance of *Verrucomicrobia* and *Deferribacteres* (Figure 5b–d). At the genus level, we found that OP treatment markedly increased the relative abundance of *Akkermansia* in *db*/*db* mice, and significantly decreased the relative abundance of *Prevotella*, *Odoribacter*, *Ruminococcus*, and *Parabacteroides* (Figure 5f–j).

### 3.6. Key Phylotypes of Gut Microbiota Responding to OP Treatment

LEfSe analysis was performed at the OTU level (LDA score > 3). OP treatment resulted in an enrichment in *Deferribacteres* and *Verrucomicrobia* at the phylum level, 4C0d_2, *Deferribacteres*, *Alphaproteobacteria*, *Epsilonproteobacteria*, and *Verrucomicrobiae* at the class level, YS2, *Deferribacterales*, *Bacillales*, *Campylobacterales*, and *Verrucomicrobiales* at the order level, *Deferribacteraceae* and *Verrucomicrobiaceae* at the family level, and *Rikenella*, *Lactococcus*, *Helicobacter*, and *Akkermansia* at the genus level (Figure 6a,b).

### 3.7. Correlation between the Gut Microbia and Diabetes-Related Indicators

Spearman’s correlation analysis was conducted to investigate the relationship between the diabetes-related indexes and the relative abundance changes of intestinal microbiota in *db*/*db* and *db*/*db* + OP group mice. At the species level, *Bacteroides acidifaciens*, *Desulfovibrio C21_c20*, and *Bacteroides eggerthii* were significantly and negatively associated with blood glucose at the time point of 15 min during OGTT; *Bacteroides plebeius* and *Akkermansia muciniphila* were significantly and negatively related to fasting blood glucose, blood glucose at the time point of 15 min during OGTT, and HOMA-IR; *Mucispirillum schaedleri* was significantly and negatively associated with fasting blood glucose and blood glucose at the time point of 15 min during OGTT; and *Bifidobacterium pseudolongum* and *Faecalibacterium prausnitzii* were significantly and positively associated with HOMA-IR. The purple and green colors showed positive and negative correlations among species, respectively (Figure 7).

### 3.8. Predicted Metabolic Profile of the Gut Microbiota after OP Treatment

To investigate the effects of OP treatment on metabolic pathways of intestinal flora in *db*/*db* mice, we carried out PICRUSt analysis. The extended error bar plot showed that the lipid metabolism and xenobiotics biodegradation and metabolism of bacteria were prone to decrease in *db*/*db* group mice compared with that in *m*/*m* group mice, while OP treatment significantly increased these metabolic pathways. The nucleotide metabolism was prone to increase in *db*/*db* group mice compared with that in *m*/*m* group mice, while OP treatment significantly restrained the metabolic pathway. OP treatment increased the metabolism of terpenoids and polyketides and glycan metabolism and biosynthesis and decreased enzyme families. OP did not alter some functional pathways including energy metabolism, carbohydrate metabolism, the biosynthesis of other secondary metabolites, amino acid metabolism, the metabolism of other amino acids, and the metabolism of cofactors and vitamins (Figure 8).

## 4. Discussion

In an acute toxicity experiment, after 7 days of single administration of 1000 mg/kg OP to mice by intraperitoneal injection, no lethality or toxic effects were observed in the animals [17]. In addition, no side effects were observed after oral administration of 1000 mg/kg OP to mice for 7 consecutive days [18]. Therefore, we failed to determine the medium lethal dose value and toxic doses of OP, and OP was considered safe. In addition, the toxicity test of olive leaf extracts, whose main phenolic compound was OP, proved the non-toxic property of OP as well. No lethality was observed after single oral administration of 5000 mg/kg of olive leaf extract (containing 14% OP) to rats [19]. The Bonolive™, an olive leaf extract, contains 40% of olive polyphenols, of which the majority was OP. After chronic toxicity experiments with Bonolive™, no side effects could be observed even after 3 months of continuous oral administration of Bonolive™ to rats at a dose of 1000 mg/kg/day [20]. During our experiment, *db*/*db* mice showed no adverse effects after oral administration of OP at a dose of 200 mg/kg once a day for 15 weeks. This dose was equivalent to a daily intake of approximately 16.2 mg/kg human body weight (972 mg/60 kg person) according to the conversion method (body surface area normalization) recommended by the US Food and Drug Administration and Reagan-Shaw et al. [21].

Patients with a history of type 2 diabetes for more than 10 years failed to control their blood glucose well even under various interventions, and their type 2 diabetes progression was in the late stage. Some commercialized drugs, such as metformin, have been proven to be more powerful in an early stage of type 2 diabetes compared with the late stage [22,23]. Previously, studies have shown that OP has excellent performance in fighting the early stage of type 2 diabetes [11,24]. Here, our experimental results showed that OP was also very effective in combating the late stage of type 2 diabetes (Figure 1a,b). Therefore, OP may be useful in fighting type 2 diabetes throughout the whole process and could be a choice for the treatment of type 2 diabetes. Nevertheless, more preclinical or clinical trials still need to be conducted.

Although the exact relationship between gut microbes’ composition and the onset and progression of type 2 diabetes is still unclear [25], slowing down the prevalence of diabetes by adjusting the intestinal microbes’ composition is a promising method [6]. Concomitant with the improved type 2 diabetes, we observed that OP altered the microbial composition of *db*/*db* mice (Figure 5). Previous studies demonstrated that the decrease in Bacteroidetes abundance and the increase in *Verrucomicrobia* abundance were accompanied with the improvement of type 2 diabetes [26,27]. In line with this, our results showed that OP treatment significantly decreased the relative abundance of Bacteroidetes and increased the relative abundance of *Verrucomicrobia* at phylum level in *db*/*db* mice. Li et al. found that the improvement of diabetes was related to a decrease in the relative abundance of *Deferribacteres* [28]. In contrast, our results demonstrated that OP treatment markedly increased the relative abundance of *Deferribacteres*. Studies have shown that increased *Akkermansia* and decreased *Ruminococcus* and *Parabacteroides* are accompanied with the alleviation of type 2 diabetes [29,30]. In our results, OP treatment significantly increased the relative abundance of *Akkermansia* and decreased the relative abundance of *Ruminococcus* and *Parabacteroides* in *db*/*db* mice. The *Odoribacter* is known to be related to the improvement of type 2 diabetes [31]. On the contrary, our results revealed that OP treatment markedly decreased the relative abundance of *Odoribacter*. Evidence revealed that some *Prevotella* strains in the gut improve glucose metabolism, and some other strains promote diseases such as metabolic syndrome and obesity [32,33]. Our results revealed that OP treatment reduced the relative abundance of *Prevotella* in *db*/*db* mice.

*Akkermansia muciniphila* is considered to be beneficial to the host’s metabolic health and is a promising next-generation probiotic and novel food supplement [34]. In previous studies, the intervention of natural products, such as anthocyanin-rich extract of açai, rhubarb, phlorizin, and a combination of berberine and stachyose, alleviated metabolic diseases, lowered the blood glucose, and enhanced the relative abundance of *Akkermansia muciniphila* in the gut [35,36,37,38]. Similarly, in our experimental results, Spearman’s correlation analysis showed a negative correlation between the abundance of *Akkermansia muciniphila* and fasting blood glucose, blood glucose at the time point of 15 min during OGTT, and HOMA-IR (Figure 7). Therefore, the hypoglycemic effect of OP on *db*/*db* mice may be associated with its ability to increase the relative abundance of *Akkermansia muciniphila*. The decrease in blood glucose has been reported to be accompanied with an increase in the relative abundance of *Bacteroides plebeius*, *Bacteroides acidifaciens*, *Mucispirillum schaedleri*, *Desulfovibrio C21_c20*, and *Bacteroides eggerthii* in the gut [39,40,41]. Our results showed that these microbiotas were negatively correlated with the fasting blood glucose, blood glucose at the time point of 15 min during OGTT, or HOMA-IR in *db*/*db* mice (Figure 7). It has been suggested that *Faecalibacterium prausnitzii* and *Bifidobacterium pseudolongum* possess alleviating effects in type 2 diabetes [42,43]. However, our results suggested that *Faecalibacterium prausnitzii* and *Bifidobacterium pseudolongum* were positively correlated with HOMA-IR (Figure 7). More comprehensive studies are warranted to investigate the role of these microorganisms in the therapeutic efficacy of OP on the advanced stage of type 2 diabetes.

After insulin binds to the insulin receptor, the phosphorylated insulin receptor substrate 1 binds to phosphoinositide-3-kinase and activates the downstream protein Akt [44]. PTP1B, a non-transmembrane protein tyrosine phosphatase, inhibits the insulin signaling pathway via suppressing the phosphorylation of insulin receptors and insulin receptor substrates [45]. Intraperitoneal injection of the antisense oligonucleotide chain of PTP1B to *ob*/*ob* mice and *db*/*db* mice reduced the expression of PTP1B, and alleviated the type 2 diabetes of the mice [46]. Porcu et al. reported that OP treatment increased the hepatic phosphorylation of Akt in the liver and improved liver steatosis in female mice fed a high-fat diet, when the mice were orally gavaged 3% OP (dissolved in drinking water) for 8 weeks [47]. Similarly, our results showed that OP treatment inhibited the mRNA level of *PTP1B* and increased the protein level of phosphorylated Akt in the liver of *db*/*db* mice (Figure 3a,b). Thus, OP treatment may restore the impaired hepatic insulin signaling pathway in our study. In addition, our results showed that OP treatment did not change the hepatic mRNA level of *Pepck* and *G6pase* in *db*/*db* mice. Fbp1, one of the rate-limiting gluconeogenic enzymes, catalyzes fructose 1,6-diphosphate to produce fructose 6-phosphate. The inhibition of Fbp1 by its inhibitor attenuated hyperglycemia in Zucker diabetic fatty rats [48]. Our results showed that OP treatment downregulated the hepatic mRNA expression of *Fbp1* in *db*/*db* mice (Figure 3c). Based on these results, we propose that one of the mechanisms by which OP alleviates diabetes may be the activation of the insulin signaling pathway and the inhibition of the mRNA expression of *Fbp1* in the liver of *db*/*db* mice.

OP exerted ameliorative effects on diabetic complications in addition to type 2 diabetes. Untreated diabetes can cause two types of serious complications, including diabetic macroangiopathy and diabetic microangiopathy [49]. Diabetic complications are the leading cause of death in diabetic patients. One of the main features of diabetic cardiac complications is disarranged or disordered cardiac myofibrils [50,51]. A previous study reported that the administration of OP contributes to a normal histological organization in the heart cells in diabetic rats [52]. Our results revealed that OP improved the histopathological features of the cardiac tissues in *db*/*db* mice (Figure 2c). Therefore, OP may possess the potential to improve diabetic cardiac complications, and more comprehensive research deserves to be performed.

The benefits of the Mediterranean diet are well-known. The phenolic constituents present in olives and extra-virgin olive oil may be the basis for the beneficial effects of Mediterranean diets [53,54,55]. In contrast to olive leaves and drupes whose major phenolic compound is OP, extra-virgin olive oil is scarce in OP and much more abundant in its degradation product, oleacein. Oleacein has been shown to play a protective role against several metabolic abnormalities. For example, oral administration of oleacein protects mice against high-fat diet-induced adiposity, liver steatosis, and insulin resistance [56,57]. It is therefore interesting to investigate the efficacy of oleacein in preventing the advanced stage of type 2 diabetes in the future.

## 5. Conclusions

In summary, the present study demonstrated that OP administration alleviated an advanced stage of type 2 diabetes in *db*/*db* mice: it significantly decreased fasting glucose, improved glucose tolerance, and alleviated histopathological features of the pancreas, liver, and heart. The beneficial effects of OP were associated with its ability to improve the characteristics of the intestinal microbial community. These findings indicated that OP may be an effective nutraceutical that can improve advanced stage type 2 diabetes.

## Figures and Tables

**Figure 1 nutrients-13-02131-f001:**
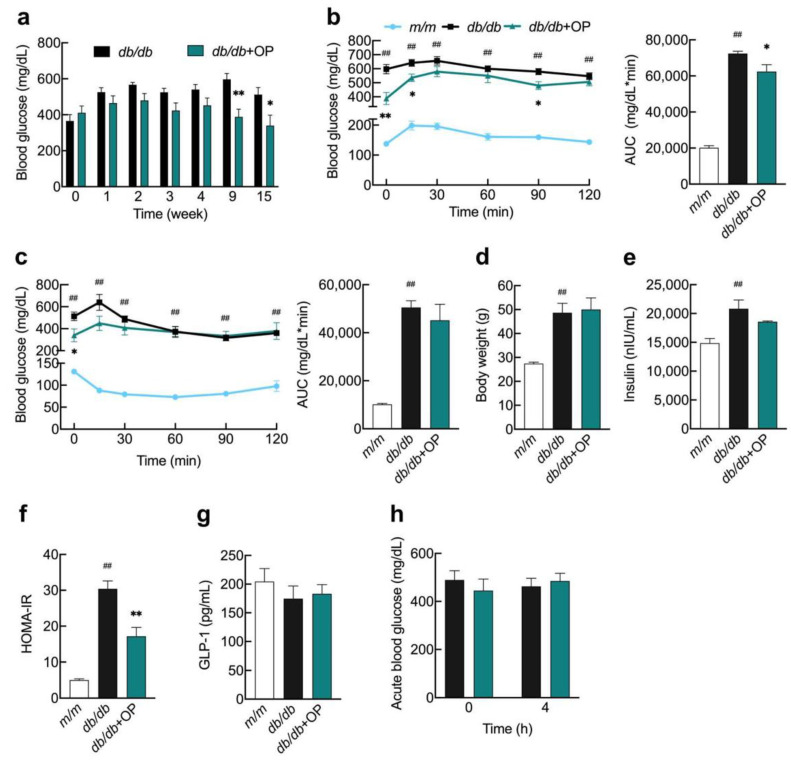
OP alleviated advanced stage of type 2 diabetes in *db*/*db* mice. (**a**) Fasting blood glucose levels of *db*/*db* group and *db*/*db* + OP group mice at week 0, 1, 2, 3, 4, 9, and 15; (**b**) Blood glucose levels during an OGTT and corresponding AUC; (**c**) Levels of blood glucose during an ITT and corresponding AUC; (**d**) Body weight; (**e**) Fasting insulin levels; (**f**) HOMA-IR index; (**g**) Fasting serum GLP-1 levels; (**h**) In *db*/*db* mice blood glucose concentrations were determined again 4 h after OP treatment. ## *p* < 0.01, *db*/*db* versus *m*/*m*. * *p* < 0.05, *db*/*db* + OP versus *db*/*db*; ** *p* < 0.01, *db*/*db* + OP versus *db*/*db.*

**Figure 2 nutrients-13-02131-f002:**
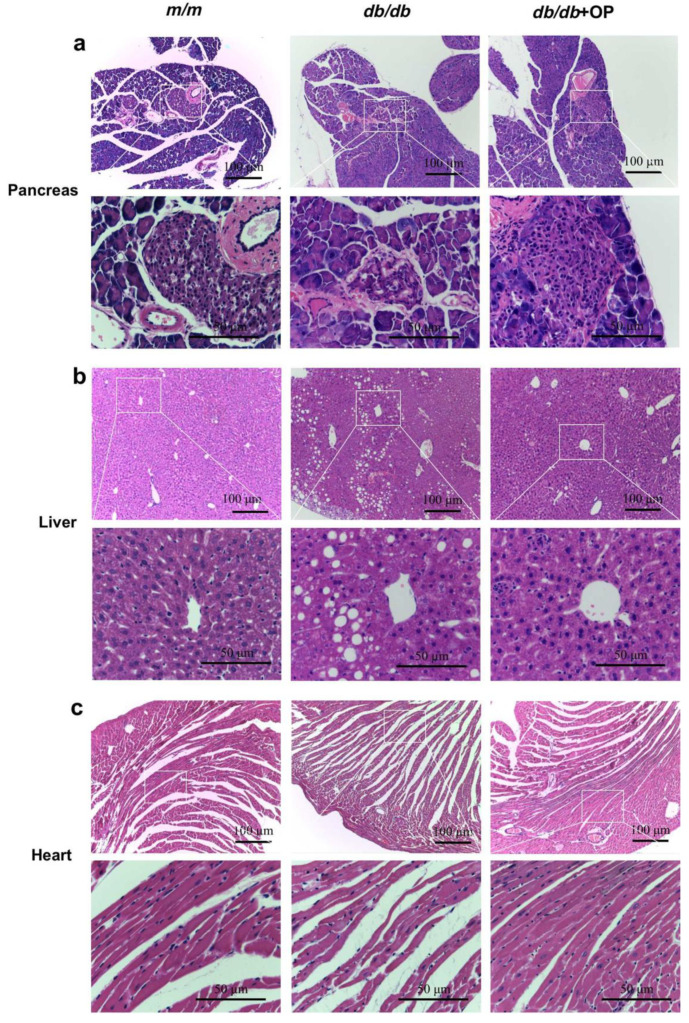
Microphotographs of H&E-stained (**a**) pancreas, (**b**) liver, and (**c**) heart from mice (100× magnification, scale bar = 100 μm; 400× magnification, scale bar = 50 μm). The rectangular frame was manually added, and the details were displayed at a higher magnification.

**Figure 3 nutrients-13-02131-f003:**
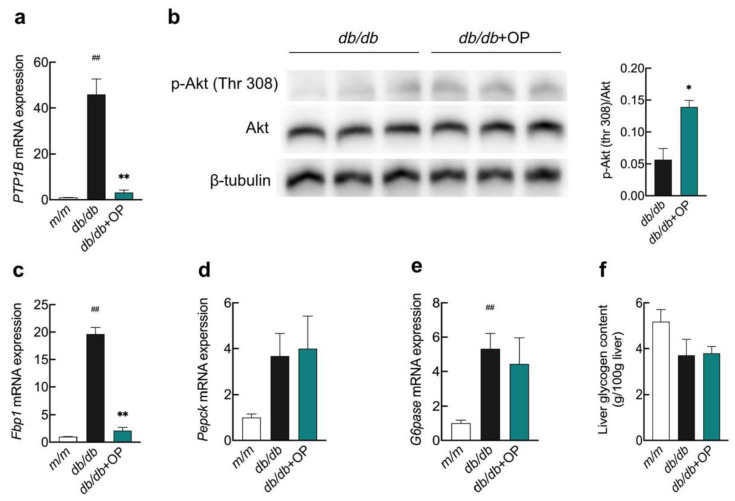
The effects of OP on hepatic insulin signaling pathway and key enzymes related to hepatic gluconeogenesis in *db*/*db* mice. (**a**) The mRNA expression of *PTP1B*; (**b**) The protein expression of p-Akt (Thr 308); (**c**–**e**) The mRNA expression of *Fbp1*, *Pepck*, and *G6pase*; (**f**) The content of liver glycogen. ## *p* < 0.01, *db*/*db* versus *m*/*m*. * *p* < 0.05, *db*/*db* + OP versus *db*/*db*; ** *p* < 0.01, *db*/*db* + OP versus *db*/*db*.

**Figure 4 nutrients-13-02131-f004:**
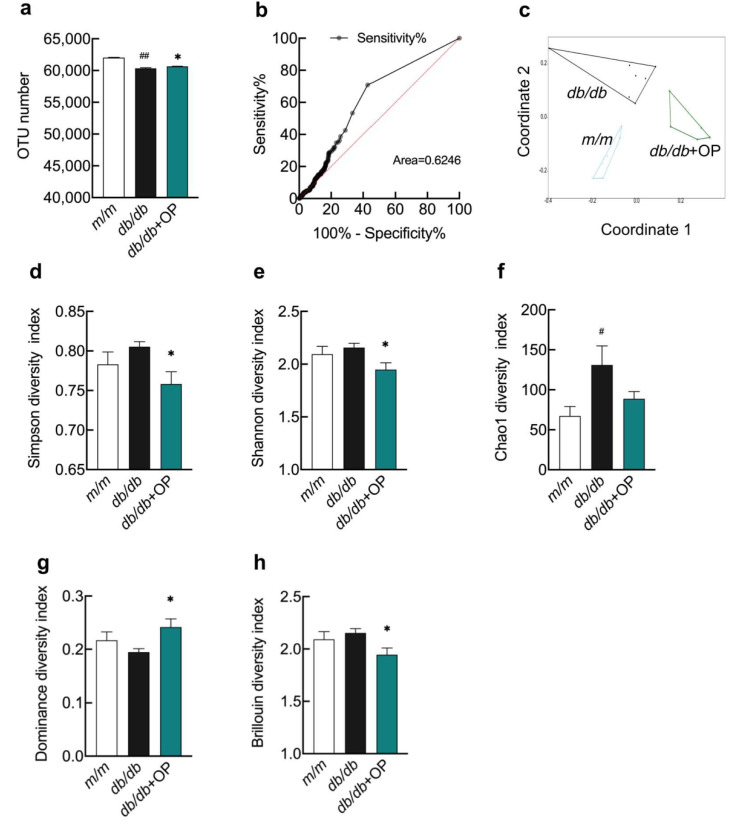
The effects of OP treatment on α-diversity and β-diversity of gut microbiota in *db*/*db* mice. (**a**) OTU numbers of *m*/*m* group mice, *db*/*db* group mice, and *db*/*db* + OP group mice; (**b**) The ROC curve; (**c**) NMDS plot based on the Bray–Curtis distances; (**d**–**h**) The α-diversity analysis, including Simpson, Shannon, Chao1, Dominance, and Brillouin diversity. # *p* < 0.05, *db*/*db* versus *m*/*m*; ## *p* < 0.01, *db*/*db* versus *m*/*m*. * *p* < 0.05, *db*/*db* + OP versus *db*/*db*.

**Figure 5 nutrients-13-02131-f005:**
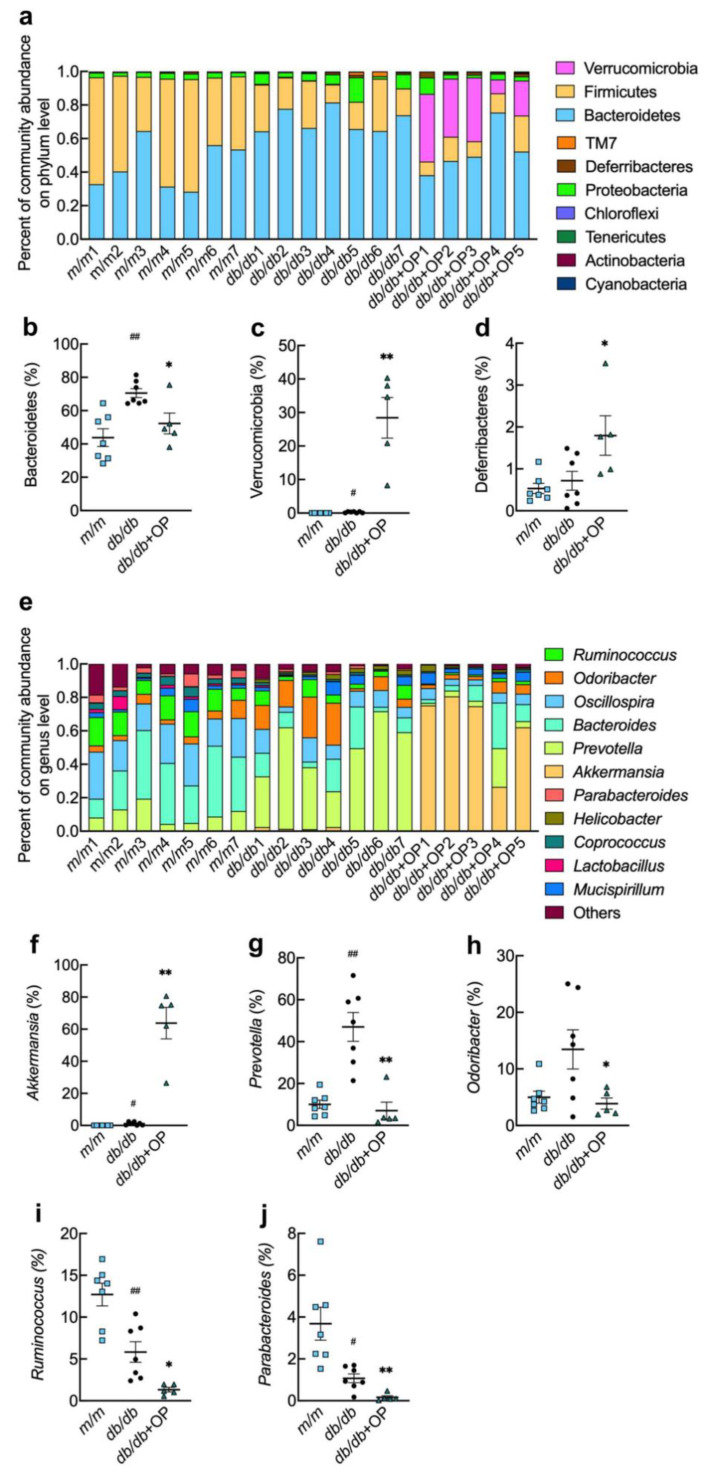
OP treatment changed the gut microbiota composition in *db*/*db* mice. (**a**) The relative abundance of intestinal microorganisms at the phylum level; (**b**–**d**) The relative abundance of *Bacteroidetes*, *Verrucomicrobia*, and *Deferribacteres*; (**e**) The relative abundance of intestinal microorganisms at the genus level; (**f**–**j**) The relative abundance of *Akkermansia*, *Prevotella*, *Odoribacter*, *Ruminococcus*, and *Parabacteroides*. # *p* < 0.05, *db*/*db* versus *m*/*m*; ## *p* < 0.01, *db*/*db* versus *m*/*m*.* *p* < 0.05, *db*/*db* + OP versus *db*/*db*; ** *p* < 0.01, *db*/*db* + OP versus *db*/*db.*

**Figure 6 nutrients-13-02131-f006:**
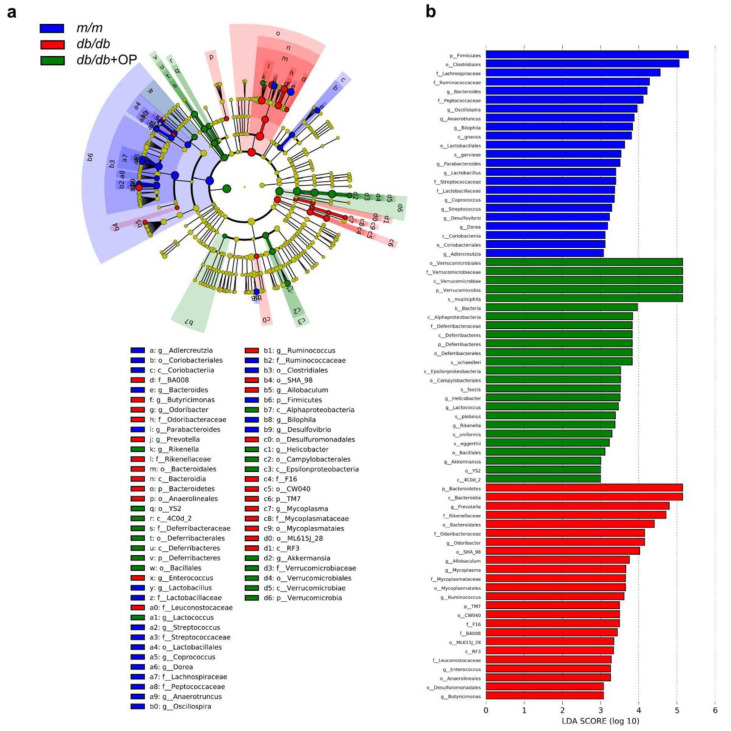
Key phylotypes of intestinal flora responding to OP treatment. (**a**) LEfSe analysis showed the relationship of OTUs (the rings, from outer to inner, represent genus, family, order, class, and phylum); (**b**) Comparison of intestinal flora among *m*/*m* group mice, *db*/*db* group mice, and *db*/*db* + OP group mice with LDA score > 3.

**Figure 7 nutrients-13-02131-f007:**
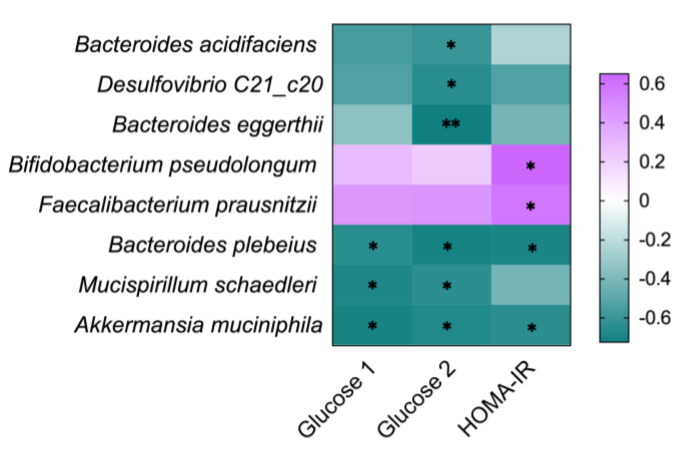
Relationship between intestinal microbiota at species level and diabetes-related indexes, including fasting blood glucose at week 9 (Glucose 1), blood glucose at time point of 15 min of OGTT (Glucose 2), and HOMA-IR. The green and purple represent negative and positive correlation, respectively. * *p* < 0.05; ** *p* < 0.01.

**Figure 8 nutrients-13-02131-f008:**
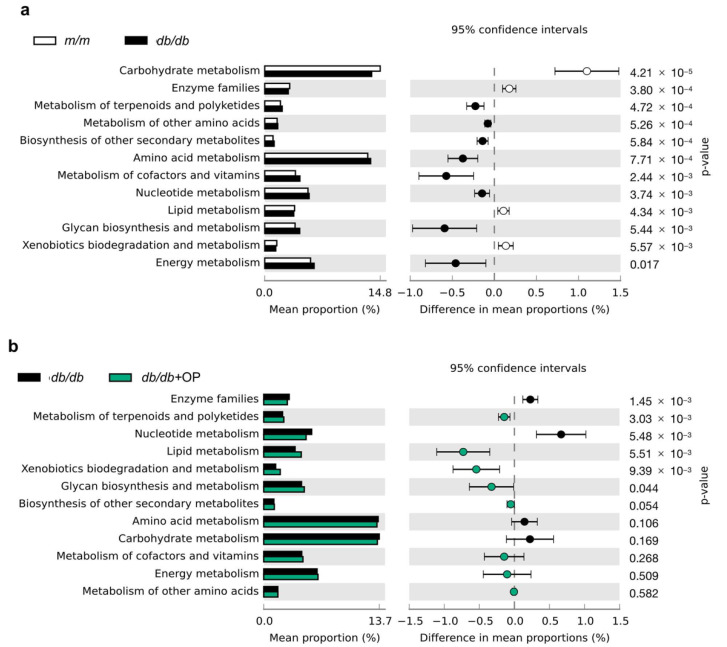
Predicated function of intestinal flora based on KEGG pathways. (**a**) The comparison of KEGG pathways between *m*/*m* group and *db*/*db* group mice; (**b**) The comparison of KEGG pathways between *db*/*db* group and *db*/*db +* OP group mice.

**Table 1 nutrients-13-02131-t001:** Primer sequences used for RT-qPCR.

Target Genes	Forward Primer 5′–3′	Reverse Primer 5′–3′
*Fbp1*	GTGTCAACTGCTTCATGCTG	GAGATACTCATTGATGGCAGGG
*G6pase*	CGACTCGCTATCTCCAAGTGA	GTTGAACCAGTCTCCGACCA
*PTP1B*	CCATCACCTCCTGGAAGAACA	TGCTGGCTTCTCTGGGTAAA
*Pepck*	CCATCACCTCCTGGAAGAACA	ACCCTCAATGGGTACTCCTTCTG
*β-actin*	GGCTGTATTCCCCTCCATCG	CCAGTTGGTAACAATGCCATGT

**Table 2 nutrients-13-02131-t002:** The effects of OP on blood lipid profiles and liver function indicators of *db*/*db* mice.

	*m*/*m*	*db*/*db*	*db*/*db* + OP
ALT (U/L)	49.60 ± 4.11	190.43 ± 38.27 #	171.90 ± 36.91
AST (U/L)	151.80 ± 24.68	282.11 ± 60.08	209.80 ± 24.68
TC (mmol/L)	2.41 ± 0.07	2.39 ± 0.34	3.39 ± 0.39
TG (mmol/L)	0.90 ± 0.06	1.28 ± 0.17	0.98 ± 0.09
LDL-C (mmol/L)	0.28 ± 0.03	0.47 ± 0.09	0.46 ± 0.08
HDL-C (mmol/L)	1.94 ± 0.06	1.96 ± 0.28	2.82 ± 0.32

# *p* < 0.05, *db*/*db* versus *m*/*m*.

**Table 3 nutrients-13-02131-t003:** Tissue mass and coefficient of liver, pancreas, and epididymal fat.

Tissues	*m*/*m*	*db*/*db*	*db*/*db* + OP
Pancreas (g)	0.30 ± 0.01	0.25 ± 0.02	0.28 ± 0.02
Pancreas %	1.08 ± 0.04	0.54 ± 0.04 ##	0.59 ± 0.09
Liver (g)	1.36 ± 0.04	2.66 ± 0.29 ##	3.04 ± 0.35
Liver %	4.98 ± 0.09	5.88 ± 0.48	6.14 ± 0.37
Epididymal fat (g)	0.28 ± 0.03	2.73 ± 0.24 ##	1.99 ± 0.36
Epididymal fat %	1.00 ± 0.08	6.00 ± 0.34 ##	3.81 ± 0.56 *

## *p* < 0.01, *db*/*db* versus *m*/*m*. * *p* < 0.05, *db*/*db* versus *db*/*db* + OP.

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
