# Peer review of "Oleuropein Ameliorates Advanced Stage of Type 2 Diabetes in db/db Mice by Regulating Gut Microbiota"

_nutrients, 2021, doi:10.3390/nu13072131_

Round 1

Reviewer 1 Report

In this manuscript, the authors described the anti-diabetes effects of oleuropein (OP) by regulating the gut microbiota. They reported OP could inhibit hyperglycemia and improve glucose tolerance. Besides, they found that OP could modulate the gut bacterial community and affect the metabolism of gut microbiota. It’s an interesting study, however, some questions need to be considered.

Comments

  1. In the manuscript, the author stated that the OP could alleviate the advanced stage of type 2 diabetes. What’s the difference between the early stage and the advanced stage? The comparison should be added in the introduction to improve the significance and novelty of this study.
  2. If possible, the results of m/m could be added in fig.1a, and fig.3 to keep all the data consistent and comparable.
  3. The gene name should be italic.

Reviewer 2 Report

The manuscript nutrients-1256970 titled “Oleuropein Ameliorates Advanced Stage of Type 2 Diabetes in db/db Mice by Regulating Gut microbiota” by Shujuan Zheng and colleagues, they have reported that oleuropein has a significant effect in decreasing fasting blood glucose levels, improving glucose tolerance, lowering homeostasis model assessment-insulin resistance index, restoring histopathological features of tissues, and promoting hepatic protein kinase B activation in db/db mice. Notably, oleuropein modulates gut microbiota at the phylum level, increases the relative abundance of Verrucomicrobia and Deferribacteres, and decreases the relative abundance of Bacteroidetes. Oleuropein treatment increases the relative abundance of Akkermansia, as well as decreases the relative abundance of Prevotella, Odoribacter, Ruminococcus, and Parabacteroides at the genus level. In conclusion, oleuropein may ameliorate the advanced stage of type 2 diabetes through modulating the composition and function of gut microbiota. I have few concerns regarding the present manuscript

-Thanks for the detailed information about materials and animals, how the dose of oleuropein was calculated, and also, how the number of animals and the weeks of treatment were calculated.

- This paper appears to use the OTU approach (though this is not adequately described in the methods). This method of clustering sequences and then assigning taxonomy to representative sequences within clusters, while not wrong, has largely been supplanted by amplicon sequence variants, such as those yielded by the DADA2 pipeline. While neither is perfect, ASVs should be better at avoiding some of the pitfalls associated with grouping different taxa based on an arbitrary % sequence identity. This could prove interesting in the developing infant gut microbiota, where Bifidobacteria make up a large percentage of the community but contain many different related taxa.

-Methods for count-based data (controlling for sequencing depth), or models of variance-stabilized counts, or Rivera-Pinto's 'balances' method might all be more appropriate for statistical tests of differences in relative abundance. Even, the Rivera-Pinto allows to the authors obtain patterns for HIV and MetS population.

-The present manuscript is well-organized and well-written, congrats to the authors

Round 2

Reviewer 1 Report

Thanks for the professional responses.

Author Response

The authors sincerely thank the reviewer for carefully reading our manuscript and providing critical comments. We believe that the reviewer’s comments have resulted in a significantly improved version of our manuscript.

Reviewer 2 Report

Thank you to the authors for clarifying my previous comments about sample size, dose, and treatment, I hope that my comments about OTU vs. ASV might be useful for them in further investigations.

Author Response

The authors sincerely thank the reviewer for carefully reading our manuscript and providing critical comments. We believe that the reviewer’s comments have resulted in a significantly improved version of our manuscript. With reviewer’s direct in the mind, authors will try to apply amplicon sequence variant for analyzing microbial communities in the subsequent studies.